# Prevalence and determinants of work-related ocular symptoms among dentists of Bangladesh: A cross-sectional study

Taslima Rafique[1,2], Fahima Nasrin Eva[1,3,4], Md. Shah Fattehullah Bhuyan[4,5],
Shourov Barua[4,6], Jyotie Malakar[4,7], Arpita Biswangree[4,8], Dilruba Alam Chowdhury[4,9],
Tasin Afrose[4,8], Parikshit Dutta[4,8], Jannatun Nahar[4,8], Disha Mony Dey[4],
Sreshtha Chowdhury[1,3,4]*, Mohammad Azmain Iktidar[1,4], Simanta Roy[1,4],
Mohammad Delwer Hossain Hawlader[1]

**1** Department of Public Health, North South University, Dhaka, Bangladesh, **2** Department of Physiology and Biochemistry, Sapporo Dental College, Uttara, Dhaka, Bangladesh, **3** Public Health Promotion and Development Society (PPDS), Dhaka, Bangladesh, **4** School of Research, Chattogram, Bangladesh, **5** Shaheed Suhrawardy Medical College, Dhaka, Bangladesh, **6** American International University - Bangladesh (AIUB), Dhaka, Bangladesh, **7** Department of Maternal and Child Health, National Institute of Preventive and Social Medicine, Dhaka, Bangladesh, **8** Chattogram Medical College, Chattogram, Bangladesh, **9** Chattogram Maa-O-Shishu Hospital Medical College, Agrabad, Chattogram, Bangladesh

* sreshtharoytuli@gmail.com

## Abstract

### Objectives

The present study aims to investigate the prevalence of work-related ocular symptoms and their associated factors among Bangladeshi dentists.

### Methods

This cross-sectional study was conducted among 747 practicing dental surgeons working in government and private facilities across Bangladesh. Participants were recruited using snowball sampling, and an online semi-structured questionnaire was used for data collection. The study collected sociodemographic information, clinical practice details, and ocular symptoms-related information. Regarding ocular symptoms, the participants were asked to report the occurrence of common ocular symptoms (e.g., eye itching, eye pain, blurring) in the last month.

### Results

The study found a high prevalence of ocular symptoms among dentists, with the most common being eye itching (46.85%), blurring of vision (41.1%), and eye pain (40.7%). Female dentists were more likely to report ocular symptoms, with males having lower odds of eye pain (AOR: 0.53, 95% CI: 0.40–0.85) and itching (AOR: 0.58, 95% CI: 0.40–0.85). Smoking was a strong predictor of eye pain (AOR: 1.33, 95% CI: 1.19–3.11) and itching (AOR: 1.92, 95% CI: 1.19–3.11). Dentists

**Data availability statement:** Data will be available upon reasonable request to the study supervisor.

**Funding:** The author(s) received no specific funding for this work.

**Competing interests:** The authors have declared no competing interests exist.

working > 28 hours per week (AOR: 1.91, 95% CI: 1.07–3.44) and attending >5 patients/day (AOR: 1.51, 95% CI: 1.08–2.12) had higher odds of developing eye pain, while routine ocular checkups were associated with lower odds of eye pain (AOR: 0.62, 95% CI: 0.44–0.89).

## Conclusion

Ocular symptoms are highly prevalent among Bangladeshi dentists, emphasizing the need for regular eye checkups, ergonomic practices, and workload adjustments to mitigate occupational risk factors and protect visual health.

## Introduction

According to the World Health Organization (WHO), ocular health embodies a holistic state of physical, social, and mental well-being concerning vision, surpassing the absence of disease and infirmity [1]. This holistic perspective highlights the importance of optimal ocular health as an essential element of overall well-being and quality of life [2]. The WHO reports that 285 million people globally live with visual impairment. Among them, 246 million have low vision, 39 million are blind, and two-thirds of this population are over 50 years [3]. Moreover, data suggest that up to 90% of all visual impairment is avoidable by prevention, treatment, or cure [4]. Hence, ocular health is increasingly recognized as a critical issue in the healthcare sector and society, as undetected and untreated ocular conditions can lead to vision loss and blindness [5]. These issues are particularly prevalent among working professionals exposed to prolonged screen time, poor lighting conditions, and inadequate ergonomic practices [6].

Dentists, in particular, face unique challenges that make them susceptible to ocular issues [7]. Good vision is crucial in dentistry, where precise clinical tasks, such as aesthetic dental work involving dentures, bridges, composites, and tooth bleaching, require high visual acuity [8]. Dentists frequently use magnification devices like loupes to enhance their vision, but these can strain their eyes over extended periods [9]. Poor illumination in dental work areas, curing lights, and trimming procedures also poses significant risks to eye health [10]. Additionally, dentists are exposed to aerosols, particulate matter, and chemical fumes, which can exacerbate ocular symptoms [11]. Moreover, individual factors like age, gender, BMI, and smoking status influence ocular symptoms, with older dentists and those with higher BMI or who smoke being more at risk [12,13]. A recent survey revealed that over 31% of dental students were unsure if their eyesight was adequate for practicing dentistry [14]. Also, studies have shown that nearly 50% of dentists report experiencing eye strain, dryness, and discomfort, exacerbated by the constant use of digital screens and environmental factors such as bright lights and reflections [15]. Moreover, prior literature suggests that regular eye examinations and routine health checkups are vital determinants for detecting potential issues early, enabling timely interventions, and preventing the progression of eye-related conditions [16].

In the Southeast Asian region, home to a quarter of the world's population, nearly one-third of the world's 39 million blind individuals reside [3]. Within Bangladesh specifically, the prevalence of ocular issues among adults is escalating, with an estimated 750,000 people affected by blindness and a significant proportion grappling with vision impairments due to a range of environmental and occupational factors [17]. Addressing ocular health among Bangladeshi dentists is particularly important, given the high patient load and demanding nature of their work [18]. Bangladeshi dentists often face extended working hours and high patient turnover [19], which can exacerbate ocular symptoms. While previous studies have explored musculoskeletal problems among Bangladeshi dentists [20,21], there is a lack of research focusing specifically on the diverse ocular symptoms they experience. Therefore, this study aims to fill this gap by exploring the prevalence of work-related ocular symptoms and their associated factors among Bangladeshi dentists, providing valuable insights for improving their occupational health and well-being.

## Methodology

### Study population and sampling

A cross-sectional study was conducted among 747 dental surgeons from government and private facilities chosen conveniently from different cities in Bangladesh. Dental surgeons who were currently practicing dentistry at least once a week and had consented electronically to participate in the study were included. We excluded foreign nationals residing in Bangladesh, people with disability, or any condition that can affect eye or vision (e.g., glaucoma, cataract).

### Sample size

We initially calculated the sample size (385) considering the sample size formula for single proportions: $n = z^2 \times p \times (1 - p)/d^2$; where: $z = 1.96$ for a confidence level of 95%, $p =$ proportion (from Alsabaani et al. (2017) [8] =51%), $d =$ margin of error $= 0.05$. Considering the 10% non-response, the final sample size was $385 + 38 = 423$.

Given that the above formula assumes a simple random sample, and we employed snowball sampling, we approached more than double (850) of the required sample size. Out of 850 participants, 747 (~88%) finally completed the survey and were considered for analysis (Fig 1).

### Survey administration

Data collection occurred over the period from January 15th to May 20th, 2024. Dental surgeons from different cities in Bangladesh were reached out through mobile phone/social media using the snowball sampling technique. The participants received a Google form link via Facebook messenger/email/SMS (making it a closed survey) containing the consent form and questionnaire. The first part of the Google form included detailed information on the aims and objectives of the research, along with a check box for consenting electronically. Only participants who consented were allowed to proceed further and complete the online semi-structured questionnaire.

### Study instrument

The questionnaire included socio demographic, clinical practice-related information, and ocular symptoms-related information. Socio demographic sections included information related to age, gender, BMI, work status, current smoking status, sleep pattern, and routine ocular checkups. Use of laser, use of light curing system, air conditioner usage in practice, number of patients treated/day, dental practice hours/week, and break between patient dealings were included in clinical practice-related information. In the last section, the participants were asked to report the occurrence of common ocular symptoms (yes or no) in the last month. The common ocular symptoms (eye itching, eye pain, blurring, feeling that eyesight is worsening, eye redness, eye burning sensation, tear, increased sensitivity to light, feeling of a foreign body, eye dryness, difficulty focusing for near vision, excessive blinking, heavy eyelids, double vision, colored halos around objects)

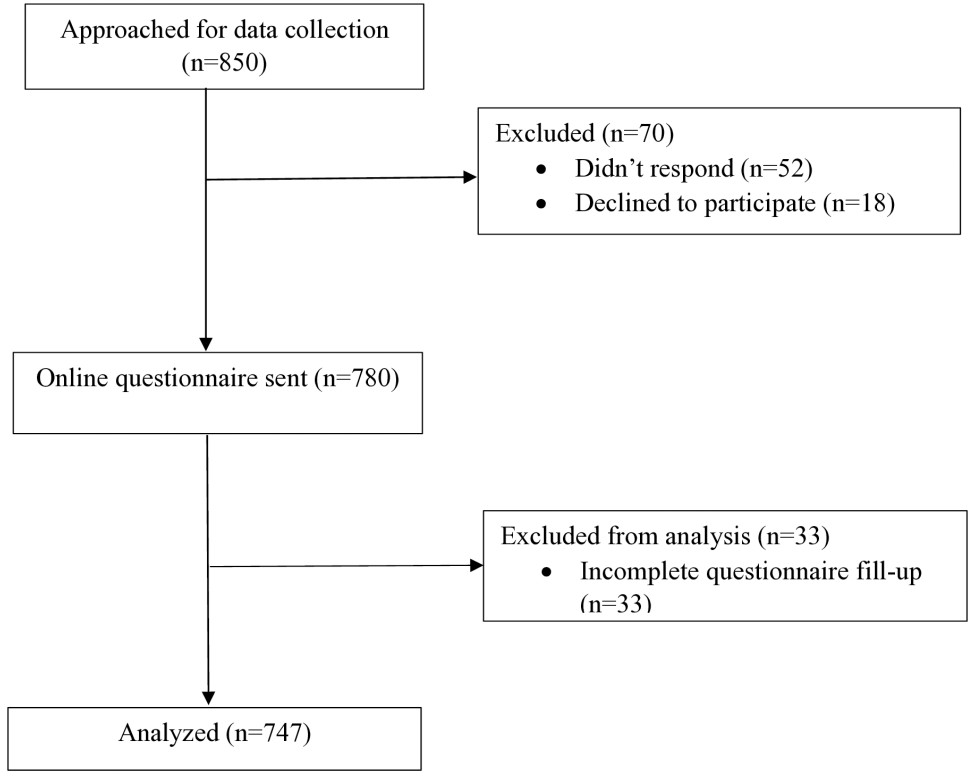

**Fig 1. Flow chart of participants at different stages of the study.**

were identified through an extensive literature review and expert opinion. The questionnaire was inputted into Google Forms without randomizing items for online distribution and tested for usability and technical functionality. Mandatory items were highlighted with a red asterisk, and the relevant non-response option was present. Respondents could review their answers through the back button and change their responses if necessary. The survey was never displayed again once the user had filled it in to prevent duplicate entries. The complete English version of the questionnaire is available in the supplementary materials (S1 File).

## Pretesting

A pilot survey preceded the main survey. The survey questionnaire was distributed to 46 (23 + 23) dental surgeons from a government and a private facility chosen at random. Participants were recruited from those institutions conveniently. The final questionnaire was prepared based on input from the pilot study. Some questions have been revised to improve their clarity and correctness of phrasing.

## Ethics

The Institutional Review Board of North South University approved (approval no: #2023/OR-NSU/IRB/1234) the research, and all participants provided written informed consent electronically. Wherever feasible, the 1964 Declaration of Helsinki and later modifications and comparable ethical standards were followed. Data collection was voluntary, and no incentives were offered to participants. Data were only accessible to the authors and were not disclosed anywhere. All the reporting

was done according to the Checklist for Reporting Results of Internet E-Surveys (CHERRIES) guidelines. The completed checklist is included in the supplementary materials.

## Statistical analysis

The Stata (version 17) was used to analyze the data, and R programming (Version 2023.12.1+402) was used for figure generation. A histogram, a normal Q–Q plot, and the Kolmogorov-Smirnov test were used to check for normality in continuous data. The categorical variables were displayed as frequencies with their respective percentage, whereas means and standard deviations were reported for continuous variables. The association between categorical independent variables (e.g., age, sex) and dependent variables (eye itching, eye pain, blurring, and feeling of eyesight worsening) was evaluated using Pearson's chi-squared test and independent sample t-test. Logistic regression models were fitted to explore the predictors of outcome variables. The explanatory variables in this study were chosen through a combination of statistical approaches. Stepwise regression alone was not used, as this approach can exclude variables of established epidemiological importance. The lowest values of the Akaike Information Criterion and the Bayesian Information Criterion (BIC) were considered while considering the model selection. The variance inflation factor (VIF) was used to measure the presence of multicollinearity (VIF < 5 for all). An interaction term between weekly dental practice hours and the break interval between patients was included to examine their joint effect on ocular symptoms. Lastly, the logistic regression analysis obtained adjusted ORs (AORs) and corresponding 95% CIs. A p-value of <0.05 was considered statistically significant.

## Results

A comprehensive analysis was conducted on a sample of 747 practicing dentists to determine the common ocular symptoms and factors associated with them. Participants' mean age and BMI were 32.16±2.5 and 24.75±2.5, respectively, with a balanced representation of male (50.47%) and female (49.53%). Most of the study participants were self-employed (35.34%), followed by 33.20% working in public organizations and 27.71% in private organizations. Nearly half of the participants (49.80%) had sound sleep, and only 17.40% of them had a current smoking habit. Most (59.17%) had their eyes checked regularly by specialists (Table 1).

Most of the study participants practiced dentistry for more than 28 hours weekly (61.42%) with no (33.73%) or 5–10 minutes break (39.22%) between patients. More than half of them treated more than five patients per day (52.10%) on average. During dental practice, 20.08% reported irregular use of laser, with 58.63% looking into the light curing unit from a safer distance. Most of the participants reported the availability of air-conditioning systems (72.22%) at the workplace.

The bar plot in Fig 2 indicates the prevalence of various ocular symptoms experienced by dentists in Bangladesh. The most common ocular problems among dentists were itching in the eyes (46.85%), blurring of vision (41.1%), eye pain (40.7%), and feeling of worsening of eyesight (39.22%). Other reported issues were eye redness (37.48%), burning sensation (37.35%), tearing (34.81%), increased sensitivity to light (33.47%), feeling of foreign body in eyes (32.26%), and dryness of eyes (30.12%). However, difficulty in focusing near vision (26.77%), excessive blinking (23.16%), heavy eyelids (21.15%), double vision (20.75%), and colored halos around objects (20.35%) were the less common ocular symptoms.

Table 2 provides insights into the bivariate relationships between various factors and eye-related symptoms among dental practitioners. Gender and routine ocular checkups were significantly associated with eye itching, whereas age and sleep patterns were significantly associated with blurring of vision. In terms of eye pain, gender, sleep pattern, use of the light curing unit, and routine ocular checkups had statistically significant relationship. Age, smoking status, break after each patient, and use of the light curing unit showed significant relationship with the feeling of worsening eyesight.

In our multivariate logistic regression model, we included all the significant potential variables in bivariate analysis. With this analysis, we incorporated the adjusted result and showed it in Table 3. Firstly, increasing age was associated with higher odds of worsening eyesight (AOR 1.02, 95% CI 1.00 to 1.04) and blurring of vision (AOR 1.04, 95% CI 1.02 to 1.06). Male gender was consistently associated with lower odds of eye pain (AOR 0.53, 95% CI 0.40 to 0.85),

**Table 1. Characteristics of the study participants (N = 747).**

| Variables | N | % |
|---|---|---|
| **Age (mean±SD)** | 32.16±2.5 | |
| **Gender** | | |
| Male | 377 | 50.47 |
| Female | 370 | 49.53 |
| **Currently smoking** | | |
| No | 617 | 82.60 |
| Yes | 130 | 17.40 |
| **Sleep pattern** | | |
| Not sufficient | 85 | 11,38 |
| Sound sleep | 372 | 49.80 |
| Average | 290 | 38.82 |
| **Work status** | | |
| Not currently practicing dentistry | 23 | 3.08 |
| Public organization | 248 | 33.20 |
| Private organization | 207 | 27.71 |
| Self-employed | 264 | 35.34 |
| Others | 5 | 0.67 |
| **Use of light curing system** | | |
| Avoiding looking directly in the light | 23 | 3.08 |
| Looking at it from a safe distance | 438 | 58.63 |
| Others | 286 | 38.29 |
| **Use of laser** | | |
| Never | 542 | 72.56 |
| Irregular | 150 | 20.08 |
| Regular | 55 | 7.36 |
| **Air conditioner usage in workplace** | | |
| No | 215 | 28.78 |
| Yes | 532 | 71.22 |
| **Body Mass Index (BMI) (mean±SD)** | 24.75±2.5 | |
| **Routine ocular checkup** | | |
| No | 305 | 40.83 |
| Yes | 442 | 59.17 |
| **Number of patients treated/day** | | |
| ≤5/day | 353 | 47.90 |
| >5/day | 384 | 52.10 |
| **Dental practice hours/week** | | |
| ≤28 hours | 245 | 38.58 |
| >28 hours | 390 | 61.42 |
| **Break after each patient** | | |
| No break time | 252 | 33.73 |
| 5–10 minute | 293 | 39.22 |
| 10–20 minute | 128 | 17.14 |
| >20 minute | 74 | 9.91 |

*N=Number of observations among study participants, SD= Standard deviation.*

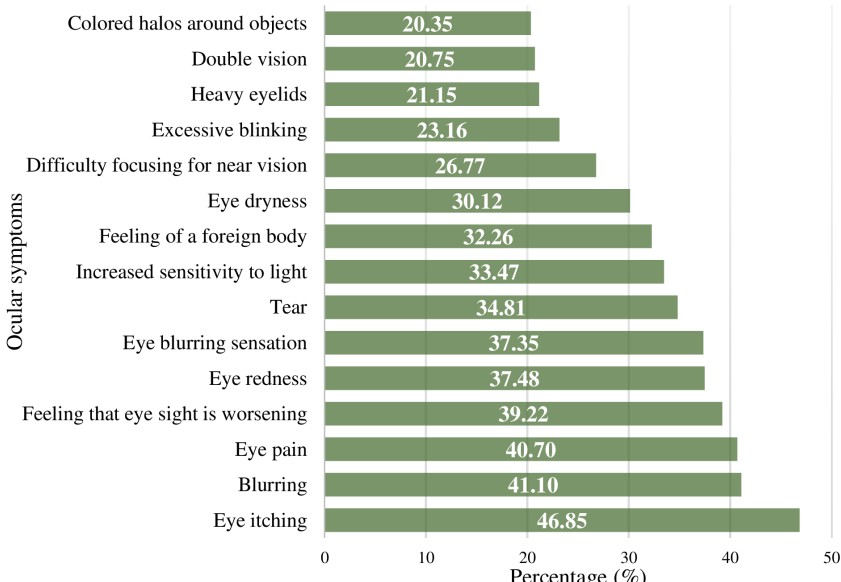

**Fig 2. Bar plot showing the prevalence of common ocular symptoms among dentists (multiple answers).**

eye itching (AOR 0.58, 95% CI 0.40 to 0.85), blurring (AOR 0.66, 95% CI 0.45 to 0.97), and the feeling that eyesight is worsening (AOR 0.61, 95% CI 0.41 to 0.89, p = 0.01) compared to females. Also, current smoking status was consistently associated with higher odds of reporting eye pain (AOR 1.33, 95% CI 1.19 to 3.11), eye itching (AOR 1.92, 95% CI 1.19 to 3.11), and the feeling that eyesight is worsening (AOR 1.88, 95% CI 1.16 to 3.06, p = 0.01) compared to non-smoking. Routine ocular checkups were associated with 38% lower odds of reporting eye pain (AOR 0.62, 95% CI 0.44 to 0.89). Furthermore, working more than 28 hours per week in dental practice was associated with higher odds of reporting eye pain (AOR 1.91, 95% CI 1.07 to 3.44). Lastly, the duration of breaks taken during dental practice also influenced the odds of reporting symptoms, with longer breaks (>20 minutes) associated with significantly lower odds of eye complaints compared to shorter breaks (AOR 0.25, 95% CI 0.07 to 0.90) and (AOR 0.23, 95% CI 0.07 to 0.85). Interaction terms were also included in the adjusted model. Dental practice hours per week had significant interactions with breaks after each patient (e.g., in terms of eye pain: >28 hours/week # 5–10 min AOR = 0.40, 95% CI: 0.18 to 0.87). While this highlights the significance seen in the interactions of these variables, the contribution of the individual terms must also be included to see the relative adjusted odds, which is achieved by multiplying the AOR estimates of the interaction term and the individual variables that make up the interaction term (e.g., >28 hours/week # 5–10 min break x >28 hours/week x 5–10 min break).

## Discussion

Eye health is vital to occupational well-being, particularly in professions where visual acuity plays a crucial role in daily tasks [2]. As primary caregivers of oral health, dentists rely heavily on their vision for precise and intricate procedures, making ocular health paramount to their professional efficacy and personal well-being [22]. They have an increased risk of ocular problems owing to prolonged exposure to dental aerosols, high-intensity operative lighting, and sustained visual demands inherent to clinical practice. Therefore, understanding the challenges associated with eye problems among dental professionals is crucial for creating specific solutions. The study identified common ocular symptoms (e.g., eye itching, blurring, pain) among dentists and factors (e.g., workload, lifestyle) associated with them.

**Table 2. Common ocular symptoms (eye itching, eye pain, blurring, feeling that eyesight is worsening) and their relationship with sociodemographic and clinical practice-related factors (N = 747).**

| Variables | Eye itching | | Eye pain | | Blurring | | Feeling that eyesight is worsening | |
|---|---|---|---|---|---|---|---|---|
| | Yes | p-value | Yes | p-value | Yes | p-value | Yes | p-value |
| **Age, mean (SD)** | 32.20(10.41) | 0.88 | 32.98(10.97) | 0.06 | 33.65(12.00) | .0.001* | 33.51(11.69) | 0.002* |
| **Gender** | | | | | | | | |
| Female | 190 | 0.02* | 169 | 0.006* | 163 | 0.10 | 152 | 0.300 |
| Male | 160 | | 135 | | 144 | | 141 | |
| **Currently smoking** | | | | | | | | |
| Yes | 282 | 0.17 | 58 | 0.32 | 59 | 0.27 | 67 | 0.002* |
| No | 68 | | 246 | | 248 | | 226 | |
| **Sleep pattern** | | | | | | | | |
| Not sufficient | 48 | 0.14 | 44 | 0.02* | 45 | 0.01* | 34 | 0.310 |
| Sound Sleep | 173 | | 124 | | 136 | | 136 | |
| Average | 129 | | 136 | | 126 | | 123 | |
| **Break after each patient** | | | | | | | | |
| No break time | 128 | 0.50 | 111 | 0.60 | 106 | 0.74 | 119 | 0.006* |
| 5–10 min. | 132 | | 113 | | 115 | | 95 | |
| 10–20 min. | 57 | | 50 | | 57 | | 49 | |
| >20 min. | 33 | | 30 | | 29 | | 30 | |
| **Work status** | | | | | | | | |
| Not currently practicing | 10 | 0.29 | 9 | 0.97 | 10 | 0.81 | 8 | 0.109 |
| Public organization | 119 | | 104 | | 106 | | 110 | |
| Private organization | 94 | | 80 | | 81 | | 73 | |
| Self employed | 127 | | 109 | | 109 | | 102 | |
| Others | 0 | | 2 | | 1 | | 0 | |
| **Use of light curing system** | | | | | | | | |
| Avoiding looking directly into the light | 9 | 0.15 | 4 | 0.02* | 8 | 0.19 | 10 | 0.002* |
| Looking at it from a distance | 218 | | 191 | | 192 | | 194 | |
| Others | 123 | | 109 | | 107 | | 89 | |
| **Routine ocular checkup** | | | | | | | | |
| No | 167 | 0.0002* | 144 | 0.003* | 133 | 0.25 | 120 | 0.960 |
| Yes | 183 | | 160 | | 174 | | 173 | |
| **Number of patients treated/day** | | | | | | | | |
| ≤5/day | 158 | 0.20 | 133 | 0.07 | 147 | 0.84 | 144 | 0.440 |
| >5/day | 190 | | 170 | | 157 | | 146 | |
| **Dental practice hours/week** | | | | | | | | |
| ≤28 hours | 120 | 0.49 | 98 | 0.34 | 106 | 0.58 | 102 | 0.970 |
| >28 hours | 180 | | 171 | | 160 | | 163 | |
| **Body Mass Index, mean (SD)** | 24.48(5.51) | 0.20 | 24.61(5.70) | 0.56 | 24.69(5.21) | 0.79 | 24.57(5.45) | 0.470 |

*=p value<0.05, N=Number of observations among study participants.

We found that 46.85% of Bangladeshi dentists reported eye itching, followed by blurring (41.1%), eye pain (40.7%), and a feeling of worsening eyesight (39.22%). The frequency of vision-related occupational health concerns among Bangladeshi dentists seems consistent with prior research on dentists worldwide [15]. However, the rates observed in

**Table 3. Factors associated with eye pain, eye itching, blurring, and feeling of worsening eyesight among dentists in Bangladesh (N = 747).**

| Variables | Eye Pain | | Eye itching | | Blurring | | Feeling that Eyesight is Worsening | |
|---|---|---|---|---|---|---|---|---|
| | AOR | 95% CI | AOR | 95% CI | AOR | 95% CI | AOR | 95% CI |
| **Age** | 1.02 | 0.99 to 1.03 | 1.0 | 0.99 to 1.03 | **1.04**** | 1.02 to 1.06 | **1.02*** | 1.00 to 1.04 |
| **Gender** | | | | | | | | |
| Female | Ref | | Ref | | Ref | | Ref | |
| Male | **0.53*** | 0.40 to 0.85 | **0.58*** | 0.40 to 0.85 | **0.66** | .45 to.97 | **0.61** | .41 to.89 |
| **Currently smoking** | | | | | | | | |
| No | Ref | | Ref | | Ref | | Ref | |
| Yes | **1.33*** | 1.19 to 3.11 | **1.92*** | 1.19 to 3.11 | 1.17 | 0.72 to 1.98 | **1.88*** | 1.16 to 3.06 |
| **Break after each patient** | | | | | | | | |
| No Break time | Ref | | Ref | | Ref | | Ref | |
| 5–10 min. | 1.29 | 0.70 to 2.53 | 1.14 | 0.61 to 2.11 | 1.17 | 0.72 to 1.91 | 0.73 | .39 to 1.38 |
| 10–20 min. | 1.27 | 0.51 to 2.41 | 0.79 | 0.37 to 1.69 | 1.17 | 0.72 to 1.91 | 1.14 | .54 to 2.44 |
| >20 min. | 2.61 | 0.52 to 3.96 | 2.51 | 0.86 to 7.31 | 1.17 | 0.72 to 1.92 | .66 | .61 to 4.51 |
| **Routine ocular checkup** | | | | | | | | |
| No | Ref | | Ref | | Ref | | Ref | |
| Yes | **0.62*** | 0.44 to 0.89 | **0.615** | 0.44 to 0.87 | 0.81 | .94 to 7.22 | 0.98 | 0.69 to 1.39 |
| **Number of patients treated/day** | | | | | | | | |
| ≤5/day | Ref | | Ref | | Ref | | Ref | |
| >5/day | **1.51*** | 1.08 to 2.12 | 1.19 | .85 to 1.66 | 0.81 | .94 to 7.22 | 0.87 | .61 to 1.22 |
| **Dental practice hours/week** | | | | | | | | |
| ≤28 hours | Ref | | Ref | | Ref | | Ref | |
| >28 hours | **1.91*** | 1.07 to 3.44 | 1.13 | .64 to 2.00 | 1.24 | .69 to 2.21 | 1.51 | .85 to 2.69 |
| **Dental practice hours/week # Break after each patient** | | | | | | | | |
| >28 hours/Week #<5–10 min | Ref | | Ref | | Ref | | Ref | |
| >28 hours/week # 5–10 min | **0.40*** | 0.18 to 0.87 | 0.69 | .32 to 1.49 | 0.65 | 0.28 to 1.44 | 0.64 | 0.29 to 1.41 |
| >28 hours/Week # 10–20 min | 0.50 | 0.19 to1.34 | 1.13 | .43 to 2.96 | 0.95 | 0.36 to 2.51 | 0.52 | 0.20 to 1.37 |
| >28 hours/Week #>20 min | 0.54 | 0.16 to 1.85 | **0.25*** | .070 to.90 | **0.23*** | 0.07 to 0.85 | 0.322 | 0.09 to 1.10 |

*= p value<0.05, **= p value<0.001, #= interaction; AOR = Adjusted odds ratio, CI = Confidence interval, N = Number of observations among study participants.

this study seem slightly higher than those reported in developed and developing nations in previous studies [2,23]. This difference could be attributed to several factors, including variations in workplace environments, ergonomic practices, and poor healthcare infrastructure in Bangladesh [24]. Also, several socio demographic determinants are notably associated with ocular issues among Bangladeshi dentists. Age, for example, could be a significant factor, with older participants displaying higher risks of worsening eyesight and blurred vision, echoing established findings on age-related ocular changes [25,26]. Dentists, who often engage in prolonged periods of focused work under artificial lighting, are particularly susceptible to these ocular symptoms [10]. This vulnerability can be attributed to various biological changes associated with aging, including diminished lens flexibility and retinal degeneration [27]. Diminished lens flexibility may impede focusing abilities, while retinal degeneration can affect visual acuity [27]. Additionally, dentists may face an increased risk of conditions such as dry eye syndrome due to reduced blinking during intricate dental procedures [28]. Recognizing these nuanced associations is imperative for developing tailored interventions aimed at safeguarding the eye health of dentists as they progress in age.

Males had a lower prevalence of different eye complaints than females, indicating a substantial sex-based difference in eye health outcomes. Prior studies support these findings, emphasizing that variations in ocular health outcomes may arise from many factors, including biological variations, behavioral and perceptual nuances, cultural and societal influences, diverse lifestyle patterns, and disparities in healthcare-seeking behavior between genders [29,30]. In addition, another study examines the profound effects of hormonal changes in females, namely those linked to menstruation, pregnancy, and menopause, on eye health. These changes may lead to illnesses like dry eye syndrome and age-related macular degeneration [31]. Recognizing and addressing these complex factors is imperative for devising comprehensive strategies to advance gender-inclusive approaches to promoting eye health equity.

Furthermore, current smoking status, which is a lifestyle factor, is a crucial factor significantly affecting the prevalence of different eye problems among dentists in this study. The study's consistent results about the association between current smoking and higher chances of reporting eye pain, itching, and poor eyesight are aligned with the findings of previous research [32]. Numerous studies have demonstrated that smoking increases the risk of various ocular conditions, including cataracts, age-related macular degeneration, dry eye syndrome, and optic nerve damage [33,34]. Smoking is thought to exacerbate these conditions through multiple mechanisms, including oxidative stress, inflammation, vascular dysfunction, and impaired tear production [35]. Furthermore, nicotine and tar in cigarettes can directly damage ocular tissues and compromise visual function [33]. The results highlight the critical need for smoking cessation interventions and personal awareness to decrease eye-related issues from smoking.

Our study found that dentists with regular ocular checkups reported fewer ocular issues, supporting previous research reinforcing the benefits of early detection and timely intervention in ocular conditions [36]. Routine eye examinations facilitate the identification of underlying eye conditions, such as refractive errors, dry eye syndrome, and ocular surface disorders, before they progress to severe stages [37]. Moreover, regular checkups enable optometrists and ophthalmologists to monitor changes in ocular health over time, implement appropriate interventions, and educate patients on eye care practices [38]. By emphasizing the role of routine ocular checkups in preventive eye care, healthcare professionals can empower individuals to prioritize their eye health and reduce the risk of experiencing eye-related symptoms and complications.

Dental practice-related aspects, including working over 28 hours per week, are significantly associated with eye pain, highlighting the adverse effects of extended dental practice [10,39]. Factors contributing to eye strain in dental practice include protracted periods of focusing on small details, exposure to bright overhead lights, and limited opportunities for eye rest [10]. Moreover, longer gaps between appointments with patients are linked with significantly reduced prevalence of eye complaints than shorter breaks. This emphasizes the need for proper rest periods in reducing ocular strain, supporting recommendations for frequent breaks to minimize occupational stress [40]. Moreover, the interaction effects between working hours and break duration offer further insights into the complex interplay of these variables, emphasizing the need for tailored interventions to address ocular health concerns based on individual work schedules and break patterns.

## Strengths and limitations

The study included a detailed investigation of ocular symptoms in a broad and representative sample by including dentists from government and private institutions in multiple locations. Additionally, extensive pretesting of the survey instrument ensures data clarity and validity. However, this study has several limitations. The cross-sectional design of this study precludes causal relationships, and snowball sampling may also induce selection bias, reducing research population representativeness and generalizability. To minimize this issue, we included a large sample from diverse sites and characteristics. Also, to ensure accuracy, it is recommended that ophthalmologists perform ocular assessments rather than relying on self-reporting. Given the familiarity of the study population (dentists) with medical terminology and symptoms, we anticipate minimal reporting bias regarding ocular symptoms since no clinical diagnosis was involved. Despite limitations, this is the first study (to our best knowledge) to assess the effect of dental practice on ocular symptoms, and the findings of the current study address the existing literature gap in this area and recommend further longitudinal studies to determine the long-term consequences.

## Conclusion

This study highlights the substantial impact of work-related ocular problems on dentists in Bangladesh. Our research findings indicate varying prevalence of various ocular symptoms among dentists. Extended practice hours and shorter intervals between patients and females were found to be associated with varying ocular symptoms. Future research can aid in evaluating the enduring effects of these symptoms and developing interventions aimed at mitigating ocular complications.

- **What is already known on this topic-** Ocular health is a growing occupational concern among dentists globally, with known risks including prolonged screen time, poor lighting, chemical exposure, and ergonomic strain, yet it remains understudied in specific regional contexts like Bangladesh.

- **What this study adds-** This study highlights the high prevalence of ocular symptoms (itching, blurring, and eye pain) among Bangladeshi dentists. It identifies new associations between these symptoms and factors such as age, smoking, and extended work hours, while demonstrating that routine eye checkups and longer breaks can reduce symptoms. Additionally, it reveals gender differences in ocular health risks, with male dentists reporting lower odds of multiple symptoms.

- **How this study might affect research, practice or policy-** The findings call for improved occupational health practices, including routine eye checkups and ergonomic adjustments, to reduce the ocular symptoms burden in dental professionals.

## Supporting information

**S1 File. Research checklist.**
(DOCX)

**S2 File. Dataset.**
(XLSX)

**S3 File. Questionnaire.**
(DOCX)

## Acknowledgments

We would like to thank Kia Das Gupta, Md Feroje Ahamed, Shaila Akter Reme, Umma Afsana Keya, Aishee Saha, Farjana Mouree, Sanjida Hossain Nandini, for their dedicated work in data collection.

## Author contributions

**Conceptualization:** Taslima Rafique, Fahima Nasrin Eva, Sreshtha Chowdhury, Mohammad Azmain Iktidar, Simanta Roy, Mohammad Delwer Hossain Hawlader.

**Data curation:** Taslima Rafique, Fahima Nasrin Eva, Md. Shah Fattehullah Bhuyan, Shourov Barua, Jyotie Malakar, Arpita Biswangree, Sreshtha Chowdhury, Mohammad Azmain Iktidar, Simanta Roy.

**Formal analysis:** Taslima Rafique, Md. Shah Fattehullah Bhuyan, Shourov Barua, Sreshtha Chowdhury, Mohammad Azmain Iktidar, Simanta Roy.

**Investigation:** Taslima Rafique, Sreshtha Chowdhury, Mohammad Azmain Iktidar, Simanta Roy, Mohammad Delwer Hossain Hawlader.

**Methodology:** Taslima Rafique, Md. Shah Fattehullah Bhuyan, Jyotie Malakar, Sreshtha Chowdhury, Simanta Roy.

**Project administration:** Taslima Rafique, Sreshtha Chowdhury, Mohammad Azmain Iktidar, Simanta Roy, Mohammad Delwer Hossain Hawlader.

**Resources:** Taslima Rafique, Sreshtha Chowdhury, Mohammad Azmain Iktidar, Simanta Roy.

**Software:** Taslima Rafique, Sreshtha Chowdhury, Mohammad Azmain Iktidar, Simanta Roy.

**Supervision:** Taslima Rafique, Sreshtha Chowdhury, Mohammad Delwer Hossain Hawlader.

**Validation:** Taslima Rafique, Sreshtha Chowdhury, Mohammad Azmain Iktidar, Simanta Roy.

**Visualization:** Taslima Rafique, Sreshtha Chowdhury, Mohammad Azmain Iktidar, Simanta Roy.

**Writing – original draft:** Taslima Rafique, Fahima Nasrin Eva, Md. Shah Fattehullah Bhuyan, Shourov Barua, Jyotie Malakar, Arpita Biswangree, Dilruba Alam Chowdhury, Tasin Afrose, Parikshit Dutta, Jannatun Nahar, Disha Mony Dey, Sreshtha Chowdhury, Mohammad Azmain Iktidar, Simanta Roy.

**Writing – review & editing:** Taslima Rafique, Fahima Nasrin Eva, Sreshtha Chowdhury, Mohammad Azmain Iktidar, Simanta Roy.

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
