## [Decision Letter · Decision Letter 0]

5 Aug 2025

Dear Dr. Chowdhury,

Thank you for submitting your manuscript to PLOS ONE. After careful consideration, we feel that it has merit but does not fully meet PLOS ONE’s publication criteria as it currently stands. Therefore, we invite you to submit a revised version of the manuscript that addresses the points raised during the review process.

**Kindly address the comments of the reviewers so the manuscript can be considered for publication**

We look forward to receiving your revised manuscript.

Kind regards,

Samira Adnan

Academic Editor

PLOS ONE

Journal Requirements: 

2. In the online submission form, you indicated that [Data will be available upon reasonable request to the study supervisor].

Additional Editor Comments:

Kindy revise the manuscript according to the comments of the reviewers.

Reviewers' comments:

Reviewer's Responses to Questions

**Comments to the Author**

1. Is the manuscript technically sound, and do the data support the conclusions?

Reviewer #1: Yes

Reviewer #2: Yes

Reviewer #3: Yes

2. Has the statistical analysis been performed appropriately and rigorously?

Reviewer #1: Yes

Reviewer #2: Yes

Reviewer #3: Yes

3. Have the authors made all data underlying the findings in their manuscript fully available?

Reviewer #1: Yes

Reviewer #2: No

Reviewer #3: No

4. Is the manuscript presented in an intelligible fashion and written in standard English?

Reviewer #1: Yes

Reviewer #2: Yes

Reviewer #3: No

Reviewer #1: Dear author

In general, the manuscript is well written and organized. No minor or major corrections must be done. However, the author should go through the ''Instructions to author'' section and follow the Journal's instructions regarding the edition, proof reading and references style.

Reviewer #2: Very intriguing and well due research on the topic. he study is well-structured, with clear objectives and a comprehensive methodology. Key strengths include the large sample size, relevant findings on high prevalence of ocular symptoms, and identification of associated factors such as gender, age, smoking, and work hours. The use of multivariable regression and interaction terms adds depth to the analysis.

However, several limitations affect the manuscript’s rigor. The use of snowball sampling introduces potential selection bias, limiting generalizability. Reliance on self-reported symptoms without clinical validation raises concerns about measurement accuracy. The discussion occasionally overstates causal inferences despite the cross-sectional design. Statistical interpretations, particularly of interaction terms, lack clarity, and some confidence intervals suggest marginal significance. The writing is often repetitive, and figures need improvement. Additionally, the long author list without clarification of contributions should be addressed.

Overall, the manuscript presents important findings but requires revisions to improve clarity, avoid overinterpretation, and enhance methodological transparency before it is suitable for publication.

Reviewer #3: The study is interesting and generally well executed; however, please (i) include a backward stepwise regression analysis for the overall occurrence of ocular symptoms, (ii) report the model’s accuracy, (iii) add a complete copy of the survey questionnaire to the supplementary file, (iv) examine whether the type of institution (private vs governmental)—given your original 50 : 50 sampling target—affects the outcomes, and (v) clarify whether the reported ocular symptoms were corroborated by medical examination. Check for english erroers

**Do you want your identity to be public for this peer review?** For information about this choice, including consent withdrawal, please see our Privacy Policy

Reviewer #1: **Yes: ** Mohamed Abdulmunem Abdulateef

Reviewer #2: No

Reviewer #3: No

---

## [Author Response · Author response to Decision Letter 1]

19 Oct 2025

Thank you for the overall positive review. We have attached a point-by-point response below:

Reviewer #1: Dear author

In general, the manuscript is well written and organized. No minor or major corrections must be done. However, the author should go through the ''Instructions to author'' section and follow the Journal's instructions regarding the edition, proof reading and references style.

Response: Thank you for your valuable feedback. We have carefully reviewed the revised manuscript and ensured that it aligns with the Journal’s “Instructions to Authors,” including formatting, proofreading, and reference style.

Reviewer #2: Very intriguing and well due research on the topic. The study is well-structured, with clear objectives and a comprehensive methodology. Key strengths include the large sample size, relevant findings on high prevalence of ocular symptoms, and identification of associated factors such as gender, age, smoking, and work hours. The use of multivariable regression and interaction terms adds depth to the analysis.

However, several limitations affect the manuscript’s rigor. The use of snowball sampling introduces potential selection bias, limiting generalizability. Reliance on self-reported symptoms without clinical validation raises concerns about measurement accuracy. The discussion occasionally overstates causal inferences despite the cross-sectional design. Statistical interpretations, particularly of interaction terms, lack clarity, and some confidence intervals suggest marginal significance. The writing is often repetitive, and figures need improvement. Additionally, the long author list without clarification of contributions should be addressed.

Overall, the manuscript presents important findings but requires revisions to improve clarity, avoid overinterpretation, and enhance methodological transparency before it is suitable for publication.

Response: We appreciate the reviewer’s comments regarding potential selection bias and reliance on self-reported symptoms. We would like to note that these limitations have already been acknowledged in the manuscript’s “Limitations” section. Specifically, we discussed the potential selection bias from snowball sampling and the recommendation that ocular assessments ideally be performed by ophthalmologists rather than relying solely on self-reports. Additionally, we highlight that, given the study population’s familiarity with medical terminology, we anticipate minimal reporting bias regarding ocular symptoms. Please refer to Lines 322-332 of the revised manuscript for clarification. In the manuscript, the discussion has been refined as needed, and statistical interpretations (including interaction terms) have already been clarified (please refer to lines 239-245). Additionally, we have included a detailed author contribution statement to clarify each author’s role.

Reviewer #3: The study is interesting and generally well executed; however, please (i) include a backward stepwise regression analysis for the overall occurrence of ocular symptoms, (ii) report the model’s accuracy, (iii) add a complete copy of the survey questionnaire to the supplementary file, (iv) examine whether the type of institution (private vs governmental)—given your original 50 : 50 sampling target—affects the outcomes, and (v) clarify whether the reported ocular symptoms were corroborated by medical examination. Check for english erroers

Response: We thank the reviewer for this thoughtful suggestion.

(i) Explanatory variables in this study were chosen through a combination of statistical approaches (including stepwise regression), theoretical relevance, and prior literature. As such, we did not rely exclusively on backward stepwise regression models, since this approach alone may omit variables with strong theoretical or epidemiological importance. Please refer to lines 174- 178, page 7.

(ii) To ensure model adequacy, we compared models using the lowest AIC and BIC values. We also assessed multicollinearity with VIF (<5), which confirmed the stability of the final model. Regarding performance, we reported adjusted aORs with 95% CIs, as the primary aim of our analysis was to evaluate the strength and direction of associations rather than to develop a predictive model. We have clarified these points in the manuscript in lines 176- 180, page 7.

(iii) A complete copy of the survey questionnaire has been provided in the supplementary material of the revised manuscript.

(iv) We appreciate the reviewer’s thoughtful observation. While the 50:50 representation of government and private dentists was applied only in the pilot phase to refine the questionnaire, in the main survey, participants were recruited nationwide through snowball sampling via online platforms (e.g., social media, email, SMS). During analysis, we did include institutional type (government vs. private) in the multivariable model; however, it did not show any statistically significant results and did not materially affect the associations of interest. Given that our primary focus was on individual- and behavior-related predictors of ocular symptoms, we did not retain this variable in the final model.

(v) To ensure accuracy, it is recommended that ophthalmologists perform ocular assessments rather than relying on self-reporting. Given the familiarity of the study population (dentists) with medical terminology and symptoms, we anticipate minimal reporting bias regarding ocular symptoms since no clinical diagnosis was involved.

(vi) We have carefully revised the manuscript to correct English language errors and improve readability.

---

## [Editor Report · Decision Letter 1]

21 Oct 2025

Prevalence and determinants of work-related ocular symptoms among dentists of Bangladesh: a cross-sectional study

PONE-D-25-32756R1

Dear Dr. Chowdhury,

We’re pleased to inform you that your manuscript has been judged scientifically suitable for publication and will be formally accepted for publication once it meets all outstanding technical requirements.

Kind regards,

Samira Adnan

Academic Editor

PLOS ONE
---

## [Editor Report · Acceptance letter]

PONE-D-25-32756R1

PLOS ONE

Dear Dr. Chowdhury,

I'm pleased to inform you that your manuscript has been deemed suitable for publication in PLOS ONE. Congratulations! Your manuscript is now being handed over to our production team.

Kind regards,

on behalf of

Dr. Samira Adnan

Academic Editor

PLOS ONE